# Bevacizumab versus Ramucirumab in EGFR-Mutated Metastatic Non-Small-Cell Lung Cancer Patients: A Real-World Observational Study

**DOI:** 10.3390/cancers15030642

**Published:** 2023-01-19

**Authors:** Wen-Chien Cheng, Yi-Cheng Shen, Chieh-Lung Chen, Wei-Chih Liao, Chia-Hung Chen, Hung-Jen Chen, Chih-Yen Tu, Te-Chun Hsia

**Affiliations:** 1Department of Internal Medicine, Division of Pulmonary and Critical Care, China Medical University Hospital, Taichung 404327, Taiwan; 2School of Medicine, College of Medicine, China Medical University, Taichung 404333, Taiwan; 3Department of Life Science, National Chung Hsing University, Taichung 40227, Taiwan; 4Ph.D. Program in Translational Medicine, National Chung Hsing University, Taichung 40227, Taiwan; 5Rong Hsing Research Center for Translational Medicine, National Chung Hsing University, Taichung 40227, Taiwan

**Keywords:** non-small cell lung cancer, NSCLC, epidermal growth factor receptor, EGFR, anti-angiogenesis therapy

## Abstract

**Simple Summary:**

The addition of bevacizumab or ramucirumab to systemic therapy for EGFR-mutated non-small-cell lung cancer (NSCLC) patients provides survival benefits. No study to date has compared the efficacy and safety of these two antiangiogenic therapies (AATs). This study enrolled patients with stage IIIB to IV EGFR-mutated NSCLC who were treated with first-line EGFR-TKIs between January 2014 and May 2022. The progression-free survival (PFS) of patients who received front-line AATs combined with EGFR-TKI therapy was longer than that of patients receiving later-line AATs combined with other therapies (19.6 vs. 10.0 months, *p* < 0.001). Bevacizumab and ramucirumab did not differ in PFS (24.1 vs. 15.7 months, *p* = 0.454). No difference in overall survival (OS) was observed between front-line and later-line therapy (non-reach vs. 44.0 months, *p* = 0.261) or between the two AATs (48.6 vs. 43.0 months, *p* = 0.924). The incidence of some adverse events such as bleeding and hepatitis was higher for bevacizumab than for ramucirumab but it was not significant. The effectiveness and safety of the two AATs were similar.

**Abstract:**

The combination of bevacizumab or ramucirumab with epidermal growth factor receptor-tyrosine kinase inhibitor (EGFR-TKI) therapy, chemotherapy, or immunotherapy for non-small-cell lung cancer (NSCLC) patients with EGFR mutations could have survival benefits. However, no study, to date, has been conducted to compare the efficacy and safety of these two antiangiogenic therapies (AATs). Stage IIIB to IV EGFR-mutated NSCLC patients who received first-line EGFR-TKIs between January 2014 and May 2022 were enrolled. These patients were divided into two groups: those receiving bevacizumab and those receiving ramucirumab as a combination therapy in any line of treatment. Ninety-six patients were enrolled in this study’s final analysis. The progression-free survival (PFS) of patients who received front-line AATs combined with EGFR-TKI therapy was longer than that of patients receiving later-line AATs combined with other therapies (19.6 vs. 10.0 months, *p* < 0.001). No difference in overall survival (OS) was observed between front-line and later-line therapy (non-reach vs. 44.0 months, *p* = 0.261). Patients who received these two different AATs did not differ in PFS (24.1 vs. 15.7 months, *p* = 0.454) and OS (48.6 vs. 43.0 months, *p* = 0.924). In addition, these two AATs showed similar frequencies of the T790M mutation (43.6% vs. 38.2%; *p* = 0.645). Multivariate Cox regression analysis indicated several AAT cycles as an independent good prognostic factor in OS. The incidence of some adverse events such as bleeding and hepatitis was higher for bevacizumab than for ramucirumab but it was not significant. Front-line AAT and EGFR-TKI combination therapy improved the PFS of stage IV EGFR-mutated NSCLC patients. The effectiveness and safety of the two AATs were similar.

## 1. Introduction

Mutations in epidermal growth factor receptor (EGFR) are found in 10–20% of Western patients and 40% of Asian patients with non-small-cell lung cancer (NSCLC) [1]. The standard first-line molecular targeted therapy for advanced EGFR mutations in NSCLC patients is the EGFR-tyrosine kinase inhibitor (EGFR-TKI), which includes gefitinib, erlotinib, afatinib, dacomitinib, and osimertinib [2,3,4,5,6]. Despite the good initial response to EGFR-TKIs, acquired resistance develops following first-line treatment. The main resistance mechanism of first-generation (gefitinib or erlotinib) and second-generation (afatinib or dacomitinib) EGFR-TKIs is the T790M mutation [7]. As a result, to overcome resistance to EGFR-targeted monotherapy, a novel combination strategy may be required.

The vascular endothelial growth factor (VEGF) family of soluble protein growth factors is an important mediator of angiogenesis in the tumor microenvironment [8]. The VEGF ligand family comprises VEGF-A, VEGF-B, VEGF-C, VEGF-D, and VEGF-E, and the VEGF receptor (VEGFR) family includes the placental growth factors VEGFR-1, VEGFR-2, and VEGFR-3. VEGF-A and VEGFR-2 are important for vascular permeability, angiogenesis, and the induction of VEGF biological responses [9]. EGFR-mutant NSCLC is more VEGF dependent than EGFR wild-type NSCLC [10]. Inhibition of the VEGF and EGFR pathways could improve the antitumor efficacy [11,12] and might have the potential to overcome EGFR-TKI resistance [13].

The anti-angiogenic agents approved by the FDA for the treatment of advanced NSCLC are bevacizumab and ramucirumab, which target VEGF-A and VEGF receptor-2, respectively [14]. Recent randomized controlled trials have shown that first-generation EGFR-TKIs combined with antiangiogenic therapies (AATs) are better than EGFR-TKI-only therapies for increasing progression-free survival (PFS) [15,16,17]. Several real-world studies have also shown that first- or second-generation EGFR-TKIs combined with AATs yield better PFS outcomes in treatment-naïve EGFR-mutant NSCLC [18,19,20]. The incidence of T790M mutation was not influenced by adding AATs [17,19]. In this way, the patient will not lose the opportunity to receive osimertinib. However, no significant benefits in overall survival (OS) were found for EGFR-TKI therapy with or without AATs as first-line treatment [19,21,22].

In the REVEL study, it was shown that second-line treatment with ramucirumab plus docetaxel may improve the OS of patients with stage IV NSCLC [23]. A combination of bevacizumab and paclitaxel as a second- or third-line treatment was shown to be effective and safe for advanced non-squamous NSCLC compared to docetaxel [24]. Kashiwabara et al. indicated that the addition of bevacizumab could be a useful therapeutic strategy for EGFR-mutant NSCLC patients with oligo-progression after EGFR-TKI failure [25]. The time from randomization to progression in second-line treatment or death from any cause is defined as progression-free survival 2 (PFS-2), which may be a potential surrogate for OS [26]. In the IMpower 150 study, a combination of atezolizumab and bevacizumab plus chemotherapy significantly improved PFS and OS in patients with EGFR mutations and advanced NSCLC, including those with prior TKI failure or with liver metastasis [27]. Our previous study also reported that AATs provided survival benefits not only in front-line treatment but also in second- or later-line treatment, especially in patients harboring the EGFR mutation L858R and with pleural effusion, bone metastasis, and liver metastasis [19]. Tsai et al. also demonstrated that a combination of EGFR-TKI and bevacizumab provided overall survival benefits in patients harboring EGFR mutation L858R [20].

Bevacizumab and ramucirumab work in a similar way to stop tumor blood vessels from growing. They are usually used in combination with chemotherapy, immunotherapy, or other targeted therapies. However, their efficacy has not been compared in clinical studies of NSCLC treatment. The purpose of this real-world study is to compare the effectiveness and safety of these two AATs in controlling advanced EGFR-mutant NSCLC.

## 2. Materials and Methods

### 2.1. Eligible Patients

A total of 96 patients with advanced-stage EGFR-mutant NSCLC, who received EGFR-TKI (gefitinib, erlotinib, afatinib, dacomitinib, or osimertinib) as first-line therapy and an AAT (bevacizumab or ramucirumab) in any treatment line, were enrolled in this retrospective study. The patients’ records were reviewed at the China Medical University Hospital, a tertiary referral center in Taiwan, between January 2014 and May 2022. The study was approved by the institutional ethics committee of the relevant institution (IRB number: CMUH110-REC1-244), and informed consent was waived due to the retrospective nature of the study. We staged patients based on the American Joint Committee on Cancer, 8th edition staging system. Only stage IIIB–IV patients were enrolled in the final analysis. Patients were excluded for the following criteria: did not receive at least three cycles of AAT treatment; did not undergo re-biopsy after the progression of disease; and had insufficient data for analysis. The recorded data from electronic records included patients’ baseline data (age, sex, smoking status, Eastern Cooperative Oncology Group Performance Status (ECOG PS)), lung cancer status (tumor–node–metastasis (TNM) stage at initial diagnosis, the pattern of distant metastases, EGFR mutation subtype, T790M status after disease progression), and treatment condition (first-line treatment, subsequent treatment, treatment-related adverse events, and treatment duration).

### 2.2. Antiangiogenic Therapies

In the present study, patients received bevacizumab at a dose of 7.5 mg/kg body weight every three weeks, which was different from the bevacizumab dose in the NEJ026 and ECOG 4599 trials [16,28]. A study reported that a dose of 7.5 mg/kg body weight was as effective as a dose of 15 mg/kg body weight when used in combination with chemotherapy [29]. In the present study, the dose of ramucirumab was 5–10 mg/kg body weight every three weeks, administered intravenously; this dosage was different from the dosage and frequency of ramucirumab in the RELAY and REVEL trials [17,23]. Reduced doses were administered because bevacizumab and ramucirumab are not covered by Taiwan’s National Health Insurance program for lung cancer management. In the present study, all patients treated with at least three cycles of bevacizumab or ramucirumab as an any-line combination therapy were enrolled in the final analysis. Front-line combination therapy was defined as the first-line systemic treatment after diagnosis of stage IIIB-IV NSCLC. Later-line combination therapy was defined as any subsequent anticancer therapy administered as a treatment line after disease progression.

### 2.3. Treatment and Safety Assessment

In the present study, all patients received computed tomography (CT) examination, brain imaging such as brain CT and/or magnetic resonance imaging (MRI), and positron emission tomography to determine the baseline stage at the initial diagnosis of lung cancer. After starting EGFR-TKI therapy, all patients received a chest CT every 3–4 months to assess tumor response. Brain CT or MRI were also used to assess cancer status by clinical judgment. PFS was defined from the beginning of EGFR-TKI therapy until tumor progression was identified based on the Response Evaluation Criteria in Solid Tumors v1.1 or death or censoring at the last follow-up (30 August 2022). OS was defined as the time interval between EGFR-TKI therapy initiation and death or censoring at the last follow-up (30 August 2022). Treatment-related side effects of AATs were evaluated and graded based on the National Cancer Institute Common Terminology Criteria for Adverse Events version 4.0.

### 2.4. Statistical Analyses

The data were analyzed using MedCalc for Windows version 18.10 (MedCalc Software, Ostend, Belgium). All continuous variables with normal distributions are reported as mean ± standard deviations, whereas those with non-normal distributions are presented as medians and interquartile ranges. The differences between groups were analyzed using the *t*-test for continuous data with normal distributions and the Kruskal–Wallis test for non-normally distributed data. The categorical variables are reported as numbers and percentages and the chi-square test was used to compare the differences between the independent groups. The PFS and OS were evaluated using the Kaplan–Meier method. To identify the independent prognostic factors for OS, univariate and multivariate Cox proportional hazards regression analyses were performed. A *p*-value of <0.05 was considered statistically significant. The strength of association is reported as the HR and the associated 95% confidence interval (CI).

## 3. Results

### 3.1. Patient Baseline Characteristics

A total of 794 patients were diagnosed with EGFR-mutated stage IIIB–IV NSCLC and 447 patients received EGFR-TKI as a first-line therapy from January 2014 to May 2022 (Figure 1). Ninety-six patients received combination therapy with an AAT as a front-line or later-line treatment. All of the 96 patients were diagnosed with lung adenocarcinoma. Of these patients, 47 (49.0%) received bevacizumab, with 23 patients prescribed it as a front-line treatment and 24 patients prescribed it as a later-line treatment; 49 (51.0%) patients received ramucirumab, of whom 19 were prescribed it as a front-line treatment and 30 as a later-line treatment. Table 1 shows the baseline characteristics of the study subjects. The EGFR-mutation subtypes were EGFR exon 19 deletion (*n* = 48, 50.0%), exon 21 L858R point mutation (*n* = 44, 45.8%), and uncommon mutations (L861Q, *n* = 2; E709Q, *n* = 1; and L747P, *n* = 1). Erlotinib (44.8%) and afatinib (37.5%) were the most commonly used EGFR-TKIs. The most frequently used combination therapy was erlotinib plus ramucirumab (57.1%). There were no statistically significant differences in the clinical parameters, including age, gender, ECOG PS, smoking status, distant metastasis site, front-line or later-line AAT, or cycle of AAT, between the two groups.

### 3.2. Survival Outcome and Safety Assessment

The median follow-up time was 47.4 months (range 39.7–60.9 months); 44 of the 96 patients (45.8%) died during this period. The median PFS of patients receiving AAT as a front-line combination therapy was significantly longer than that of patients receiving AAT as a later-line combination treatment (19.6 vs. 10.0 months; *p* < 0.001; Figure 2A). There was no significant difference in the median PFS between patients receiving EGFR-TKI combined with bevacizumab and those receiving EGFR-TKI combined with ramucirumab (24.1 vs. 15.7 months, *p* = 0.454; Figure 2B). The median OS was not reached in the front-line treatment group, whereas it was 44.0 months in the later-line treatment group (*p* = 0.261; Figure 3A). No significant differences in OS between patients receiving bevacizumab and those receiving ramucirumab were observed (48.6 vs. 43.0 months, *p* = 0.924; Figure 3B). The incidences of proteinuria, hypertension, and bleeding, the most common adverse events of AATs, were 19.8%, 14.6%, and 4.2%, respectively. The incidences of adverse events were similar between systemic therapy plus bevacizumab and ramucirumab. However, the incidences of treatment-related hepatitis and bleeding were higher, although not significantly, in the EGFR-TKI plus bevacizumab group than in the EGFR-TKI plus ramucirumab group (19.1% vs. 6.1%, *p* = 0.055; 8.5% vs. 2.1%, *p* = 0.156; respectively). There were no treatment-related deaths in this study (Table 2).

### 3.3. Acquired T790M Mutation and Outcome Predictors

In accordance with the eligibility criteria, no patient had a known EGFR T790M mutation at baseline. Disease progression developed in 39 patients (39/47, 82.9%) treated with bevacizumab combination therapy and in 34 patients (34/49, 69.4%) treated with ramucirumab combination therapy. All patients underwent a re-biopsy, either a liquid or tissue biopsy, when the disease progressed, and the incidence of the T790M mutation was assessed. The rates of T790M detection were similar between the different AAT groups: bevacizumab and ramucirumab (17/39 (43.6%) vs. 13/34 (38.2%); *p* = 0.645; Figure 4). Univariate and multivariate analyses were conducted to identify good prognostic factors associated with OS. According to the multivariate analysis, patients who received eight or more cycles of AAT treatment (HR: 0.452; 95% CI: 0.24–0.85; *p* = 0.014) were an independent prognostic factor of better OS (Table 3).

## 4. Discussion

To the best of our knowledge, this is the first study to investigate the effectiveness and safety of bevacizumab and ramucirumab as treatments for advanced EGFR-mutant NSCLC. We showed that the median PFS of patients receiving EGFR-TKIs in combination with AATs as a front-line treatment was longer than that of patients receiving EGFR-TKIs only. The median PFS of the two AATs used as front-line therapies was similar. The AATs used as front-line therapy and later-line therapy resulted in a similar OS. There were also no differences in the OS between the two AATs. However, bevacizumab was associated with more incidences of adverse events than ramucirumab, but this was not a statistically significant difference. Our results also suggest that more cycles of AATs are necessary to achieve better outcomes.

In the NEJ026 study, the PFS was longer in patients receiving erlotinib plus bevacizumab than in patients receiving erlotinib plus placebo as a first-line therapy for EGFR-mutant NSCLC (16.9 vs. 13.3 months, *p* = 0.016) [16]. The RELAY study also reported that erlotinib plus ramucirumab resulted in a longer PFS period than erlotinib plus placebo in treatment-naïve EGFR-mutant NSCLC (19.4 vs. 12.4 months, *p* < 0.001) [17]. A real-world study conducted in China reported a significantly prolonged PFS with a combination of first-generation EGFR-TKIs plus bevacizumab compared to EGFR-TKI monotherapy (16.5 vs. 12.0 months; *p* = 0.001). Tsai et al. also reported that a combination of EGFR-TKI (55.6% first-generation EGFR-TKI and 44.4% second-generation EGFR-TKI) and bevacizumab significantly improved PFS compared to EGFR-TKI monotherapy (17.0 vs. 11.0 months, *p* = 0.002) [20]. In another real-world study, Hsu et al. reported that afatinib plus bevacizumab also demonstrated a longer median PFS (23.9 months) [18]. Huang et al. indicated that a combination of first- or second-generation EGFR-TKI and bevacizumab resulted in a similar longer PFS (17.1 vs. 21.6 months; *p* = 0.617) [30]. Our previous study also indicated that EGFR-TKI (66.7% first-generation EGFR-TKI and 33.3% second-generation EGFR-TKI) combined with AATs (81% bevacizumab and 19% ramucirumab) led to a longer PFS than EGFR-TKI monotherapy (18.2 vs. 10.0 months, *p* < 0.001) [19]. The present study demonstrated that front-line treatment with AATs led to a longer PFS than later-line treatment (19.6 vs. 10.0 months, *p* < 0.001). However, the PFS was not significantly different between front-line therapy with bevacizumab and front-line therapy with ramucirumab (24.1 vs. 15.7 months, *p* = 0.454).

The OS in the present study did not differ between front-line and later-line treatment with AATs, a result consistent with those of previous studies [19,20,21,22]. This outcome may be due to OS being influenced by treatment regimens used after disease progression. Several previous studies indicated that second- or later-line combination therapy with bevacizumab or ramucirumab following NSCLC progression is more effective in improving survival than chemotherapy only [23,24]. AATs in later-line regimens remain important contributors to prolonging survival. Our previous study also showed that the inclusion of AATs in any treatment line of selected patients provided survival benefits [19]. The present study found no survival difference between bevacizumab and ramucirumab in any treatment line (51.7 vs. 43.0 months, *p* = 0.954). However, several cycles of AATs seem to be necessary for achieving better outcomes, suggesting that exposure to AATs is positively correlated with survival. An exploratory analysis of the REVEL trial indicated that higher ramucirumab exposure was associated with improved clinical outcomes [31]. However, few studies have examined the bevacizumab exposure–response relationship in patients with NSCLC. Studies on metastatic colorectal cancer showed a positive relationship between survival and bevacizumab exposure [32,33]. In these studies, the frequency of AAT therapy was every three weeks. Exposure was related to clinical outcomes because of unstable drug concentration. In the RELAY trial, no association was observed between ramucirumab exposure and response, suggesting that ramucirumab at 10 mg/kg every two weeks with erlotinib was an optimal strategy [34]. In the present study, the frequency of ramucirumab or bevacizumab was every three weeks; therefore, more cycles of AATs seem to be needed to overcome the effects of unstable drug concentration.

These two AATs had a similar safety profile in the present study. However, the incidences of adverse events such as bleeding and hepatitis appear to be higher for bevacizumab than for ramucirumab, although this was not a statistically significant difference. A meta-analysis of 85 randomized controlled trials showed that bevacizumab significantly increased the risk of all-grade and high-grade pulmonary hemorrhage in lung cancer patients, but the risk of any-grade pulmonary hemorrhage was not observed in lung cancer patients receiving ramucirumab [14]. Bevacizumab, as a humanized anti-human VEGF-A monoclonal antibody, prevented VEGF binding to VEGFR-1 and VEGFR-2. However, ramucirumab was another monoclonal antibody only targeting VEGFR-2. Although VEGFR-2 was the main mediator of VEGF-A-induced tumor angiogenesis [9], the high expression of VEGFR-1 on tumor cells was also associated with high tumor angiogenesis [35]. Based on these reasons, bevacizumab might lead to an increased risk of bleeding by targeting both VEGFR-1 and VEGFR-2. The incidences of hepatitis were higher in patients receiving bevacizumab than in those treated with ramucirumab, which may be explained by more patients combined with gefitinib and bevacizumab. Gefitinib frequently induces liver damage in patients with lung adenocarcinoma after chemotherapy [36]. The frequency and dose of ramucirumab in the present study were lower than those used in a clinical trial [17], which explains the lower incidences of adverse effects (such as bleeding and hepatitis) in patients who received ramucirumab than in those who received bevacizumab. Kanbayashi et al. indicated that hypertension, the number of cycles, and calcium channel blocker use were independent risk factors of proteinuria development after AATs [37]. The incidences of proteinuria and hypertension were similar for bevacizumab and ramucirumab in the present study, which can be explained by the similar cycles of these two AATs.

In the RELAY trial, the acquired EGFR-T790M mutation rate was not influenced by the addition of ramucirumab to erlotinib. T790M was observed in 43% of patients with disease progression receiving ramucirumab combined with erlotinib and in 47% of the erlotinib-only group [17]. In the NEJ026 trial, the T790M detection rate was also not influenced by the addition of bevacizumab to erlotinib. The detection rates of the T790M mutation were 24.2% in the bevacizumab plus erlotinib group and 26.1% in the erlotinib-only group [38]. Our previous study found that the addition of antiangiogenic drugs to any treatment line did not affect the frequency of T790M (38.5% to 40.6%) [19]. The differences in the T790M detection rates may be due to the differences in the sensitivities of the test methods. In the present study, bevacizumab and ramucirumab treatments showed similar frequencies of the T790M mutation. This is the first study to evaluate the differences in the detection of the T790M mutation between different AATs.

There were some limitations in the present study. First, the total number of patients was relatively small; therefore, our results should be confirmed by multicenter studies with more patients. Second, the bevacizumab and ramucirumab frequencies and doses were much lower in this study compared to the standard dose in clinical trials because AATs, especially ramucirumab, are not covered by health insurance in Taiwan. This might have led to financial difficulties in continuing with the therapies. As a result, there was a shorter PFS and lower incidence of side effects, although not significantly, for ramucirumab than for bevacizumab. Despite these limitations, the real-world effectiveness of bevacizumab and ramucirumab is similar to their efficacy in clinical trials [16,17].

## 5. Conclusions

The present study showed that the addition of bevacizumab or ramucirumab to an EGFR-TKI as a front-line treatment prolonged PFS more than their addition as later-line therapies. Bevacizumab and ramucirumab increased PFS to a similar extent and their effects on OS were also similar in all treatment lines. In addition, these two AATs had comparable safety profiles. However, large studies are necessary to confirm these findings.

## Figures and Tables

**Figure 1 cancers-15-00642-f001:**
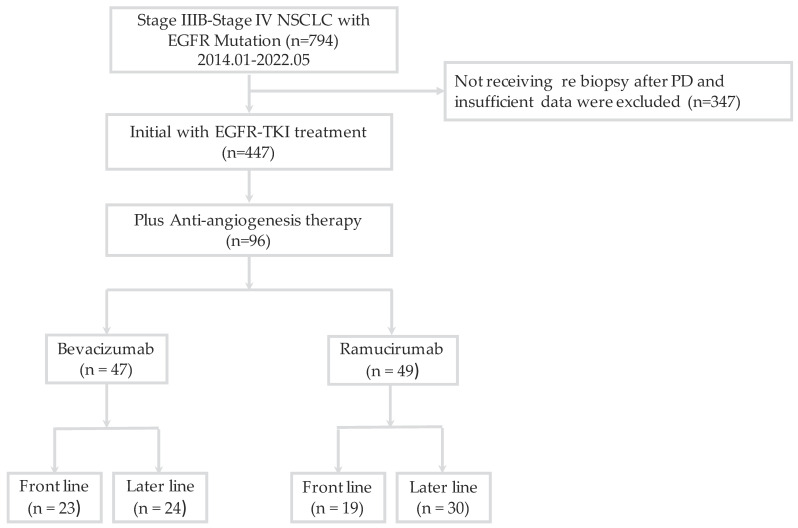
Flowchart of patient enrollment. EGFR, epidermal growth factor receptor; TKI, tyrosine kinase inhibitor; NSCLC, non-small-cell lung cancer; PD, progressive disease.

**Figure 2 cancers-15-00642-f002:**
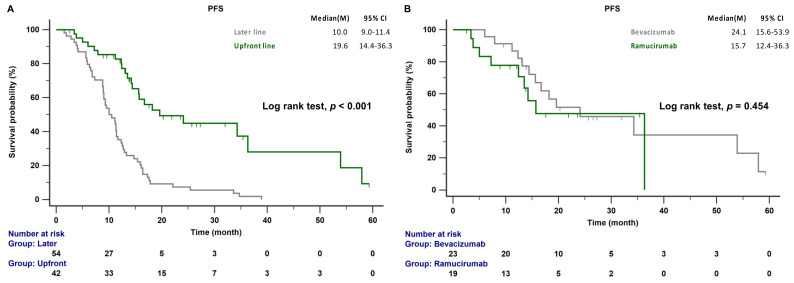
(**A**) PFS of patients with EGFR-mutant NSCLC receiving front-line and later-line therapy. (**B**) PFS of patients with EGFR-mutant NSCLC treated with front-line bevacizumab or ramucirumab combination therapy. EGFR, epidermal growth factor receptor; PFS, progression-free survival.

**Figure 3 cancers-15-00642-f003:**
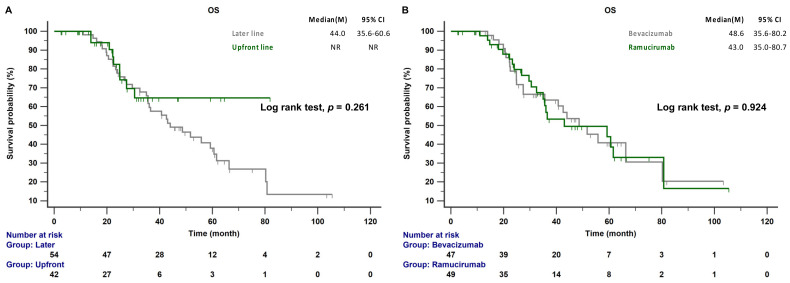
(**A**) OS of patients with EGFR-mutant NSCLC receiving front-line and later-line therapy. (**B**) OS of patients with EGFR-mutant NSCLC treated with any-line bevacizumab or ramucirumab combination therapy. CI: confidence interval; EGFR, epidermal growth factor receptor; OS, overall survival.

**Figure 4 cancers-15-00642-f004:**
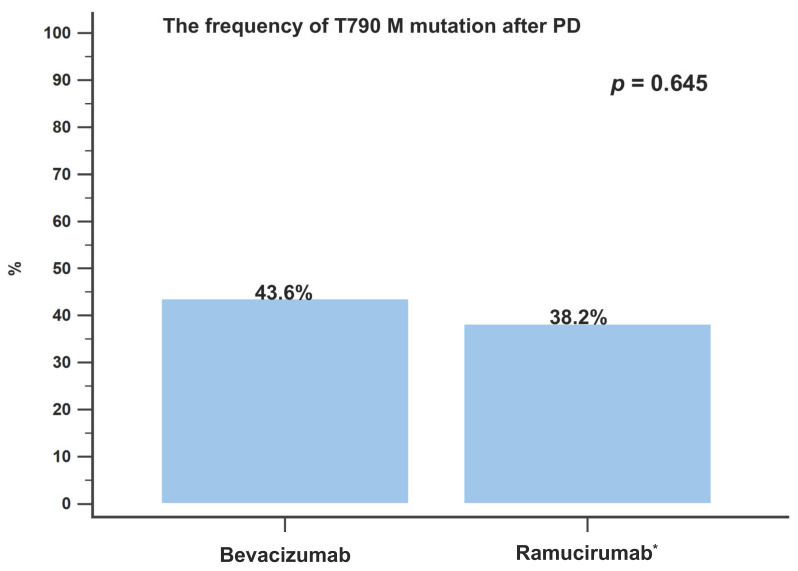
The incidence of T790M mutation development after disease progression in patients treated with bevacizumab or ramucirumab combination therapy. PD, progressive disease. * Patients treated with osimertinib were excluded.

**Table 1 cancers-15-00642-t001:** Characteristics of EGFR-mutated metastatic non-small-cell lung cancer patients receiving bevacizumab or ramucirumab in combination with epidermal growth factor receptor-tyrosine kinase inhibitor therapy.

	All (*n* = 96)	Bevacizumab (*n* = 47)	Ramucirumab (*n* = 49)	*p*-Value
Age ≥ 65 years	22 (22.9)	14 (29.8)	8 (16.3)	0.118
Male	36 (37.5)	16 (34.0)	20 (40.8)	0.495
Smoking	24 (25.0)	9 (19.1)	15 (30.6)	0.197
ECOG PS ≥ 2	5 (5.2)	2 (4.3)	3 (6.1)	0.682
*EGFR* mutation				0.394
Del 19	48 (50.0)	25 (53.2)	23 (46.9)	
L858R	44 (45.8)	19 (40.4)	25 (51.0)	
Uncommon	4 (4.2)	3 (6.4)	1 (2.0)	
Metastasis organ				
Lung metastasis	50 (52.1)	26 (55.3)	24 (49.0)	0.536
LN metastasis	65 (67.7)	33 (70.2)	32 (65.3)	0.609
Pleural metastasis	37 (38.5)	19 (40.4)	18 (36.7)	0.711
Liver metastasis	20 (20.8)	11 (23.4)	9 (18.4)	0.545
Bone Metastasis	55 (57.3)	25 (53.2)	30 (61.2)	0.428
CNS metastasis	20 (20.8)	9 (19.1)	11 (22.4)	0.692
Adrenal metastasis	7 (7.3)	4 (8.5)	3 (6.1)	0.654
Meta sites ≥ 3	35 (36.5)	17 (36.2)	18 (36.7)	0.954
EGFR-TKI				0.028
Gefitinib	13 (13.5)	9 (19.1)	4 (8.2)	
Erlotinib	43 (44.8)	15 (31.9)	28 (57.1)	
Afatinib	36 (37.5)	21 (44.7)	15 (30.6)	
Dacomitinib	2 (2.1)	2 (4.3)	0 (0)	
Osimertinib	2 (2.1)	0 (0)	2 (4.1)	
Anti-VEGF				
Front line	42 (43.7)	23 (48.9)	19 (38.8)	0.318
cycles		18 (12.3–32.1)	15 (5.8–21.2)	0.149
Later line	54 (56.2)	24 (51.1)	30 (61.2)	0.318
cycles		10 (6–14.2)	8.0 (8.0–15.0)	0.668

CNS: central nervous system; ECOG PS: Eastern Cooperative Oncology Group Performance Status; EGFR: epidermal growth factor receptor; LN: lymph node; TKI: tyrosine kinase inhibitor; VEGF: vascular endothelial growth factor. Continuous variables are presented as the means (standard deviations) or medians (interquartile ranges); categorical variables are presented as numbers and percentages.

**Table 2 cancers-15-00642-t002:** Adverse events in EGFR-mutated metastatic non-small-cell lung cancer patients receiving bevacizumab or ramucirumab in combination with epidermal growth factor receptor-tyrosine kinase inhibitor therapy.

	All (*n* = 96)	Bevacizumab (*n* = 47)	Ramucirumab (*n* = 49)	*p*-Value
Hepatitis, all grades	12 (12.5)	9 (19.1)	3 (6.1)	0.055
≥ Grade 3	1 (1.0)	0 (0)	1 (2.0)	0.327
Diarrhea, all grades	42 (46.1)	20 (42.6)	22 (44.9)	0.817
≥ Grade 3	2 (2.1)	1 (2.0)	1 (2.0)	0.976
Skin toxicity, all grades	49 (51.0)	29 (61.7)	29 (59.2)	0.802
≥ Grade 3	11 (11.5)	6 (12.8)	7 (14.3)	0.828
Paronychia, all grades	38 (39.6)	19 (40.4)	19 (38.8)	0.869
≥ Grade 3	3 (3.1)	1 (2.1)	2 (4.1)	0.584
Oral ulcer, all grades	11 (11.5)	4 (8.5)	7 (14.3)	0.377
≥ Grade 3	0 (0)	0 (0)	0 (0)	0.838
Proteinuria, all grades	19 (19.8)	11 (23.4)	8 (16.3)	0.386
≥ Grade 3	2 (2.1)	1 (2.0)	1 (2.1)	0.976
Hypertension, all grades	14 (14.6)	7 (14.9)	7 (14.3)	0.933
≥ Grade 3	2 (2.1)	1 (2.0)	1 (2.1)	0.976
Bleeding, all grades	4 (4.2)	4 (8.5)	1 (2)	0.156
≥ Grade 3	0 (0)	0 (0)	0 (0)	0.838

**Table 3 cancers-15-00642-t003:** Univariate and multivariate analyses of clinical factors associated with overall survival of EGFR-mutated metastatic non-small-cell lung cancer patients receiving bevacizumab or ramucirumab in combination with epidermal growth factor receptor-tyrosine kinase inhibitor therapy as first-line or later-line treatment.

	Univariate			Multivariate Model		
	HR	95% CI	*p*-Value	HR	95% CI	*p*-Value
Age ≥ 65 years	1.612	0.82–3.15	0.163	1.582	0.76–3.28	0.217
Ramucirumab vs. Bevacizumab	1.029	0.57–1.86	0.924	0.969	0.52–1.81	0.921
Front-line vs. Later-line treatment	0.656	0.33–1.37	0.265	1.067	0.46–2.46	0.879
Cycles of anti-VEGF ≥ 8	0.425	0.24–0.77	0.004	0.452	0.24–0.85	0.014
T790M mutation positive	0.833	0.44–1.54	0.564	0.911	0.48–1.72	0.774

CI: confidence interval; HR: hazard ratio; VEGF: vascular endothelial growth factor.

## Data Availability

The datasets used and analyzed during the present study are available from the corresponding author on reasonable request.

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
