# Peer review of "Bevacizumab versus Ramucirumab in EGFR-Mutated Metastatic Non-Small-Cell Lung Cancer Patients: A Real-World Observational Study"

_cancers, 2023, doi:10.3390/cancers15030642_

Round 1

Reviewer 1 Report

Dear Authors

This manuscript is a clinical research that conducts a long-term observation and analysis of the clinical effects and side effects of advanced NSCLC with EGFR mutation in the combined use of new or old angiogenesis inhibitory monoclonal antibody drugs and TKIs. The research is complete and the conclusions echo the results. One query needs to answer and a few text errors to correct:

1. There is one more pause in line 59 of the second paragraph of the introduction, which can be deleted.

2. There is a space missing between two sentences in the 68th line of the third paragraph of the introduction, please add it.

3. On page 5, lines 193 to 194, the repeated description of “systemic therapy plus” is recommended to be deleted.

4. NSCLC is divided into adenocarcinoma, squamous cell carcinoma, and large cell carcinoma. It is recommended to describe which type of cancer the patient belongs to when he is first diagnosed, and whether there is any difference in the content of this study.

Reviewer 2 Report

The authors conclusively presented data, and I endorse this study publication with 2 minor suggestions.

1) Authors must include information on EGFR mutations reported in NSCLC in the introduction. For example, the details on mutant T790M are missing, whereas mutant L858 was included in the introduction. That too when the L858 has no relevance with the results presented. Authors must cite the original work for the T790M mutant in the introduction.

2) In discussion, the therapy side effects should be discussed in detail with the other reports, mainly if the therapy duration is involved in these minor side effects.

Reviewer 3 Report

good work, and good luck forward. i have no comments
